# Influential Metrics Estimation and Dynamic Frequency Selection Based on Two-Dimensional Mapping for JPEG-Reversible Data Hiding

**DOI:** 10.3390/e26040301

**Published:** 2024-03-29

**Authors:** Haiyong Wang, Chentao Lu

**Affiliations:** 1School of Computer Science, Nanjing University of Posts and Telecommunications, Nanjing 210023, China; 1222045733@njupt.edu.cn; 2National Key Laboratory of Autonomous Marine Vehicle Technology, Harbin Engineering University, Harbin 150001, China; 3Smart Campus Research Centre, Information Construction and Management Office, Nanjing University of Posts and Telecommunications, Nanjing 210023, China

**Keywords:** RDH, JPEG, 2D mapping, HS

## Abstract

JPEG Reversible Data Hiding (RDH) is a method designed to extract hidden data from a marked image and perfectly restore the image to its original JPEG form. However, while existing RDH methods adaptively manage the visual distortion caused by embedded data, they often neglect the concurrent increase in file size. In rectifying this oversight, we have designed a new JPEG RDH scheme that addresses all influential metrics during the embedding phase and a dynamic frequency selection strategy with recoverable frequency order after data embedding. The process initiates with a pre-processing phase of blocks and the subsequent selection of frequencies. Utilizing a two-dimensional (2D) mapping strategy, we then compute the visual distortion and file size increment (FSI) for each image block by examining non-zero alternating current (AC) coefficient pairs (NZACPs) and their corresponding run lengths. Finally, we select appropriate block groups based on the influential metrics of each block group and proceed with data embedding by 2D histogram shifting (HS). Extensive experimentation demonstrates how our method’s efficiently and consistently outperformed existing techniques with a superior peak signal-to-noise Ratio (PSNR) and optimized FSI.

## 1. Introduction

Reversible Data Hiding (RDH) involves embedding a message into a cover image with the objective of minimizing visual distortion and enabling the lossless recovery of the original image from the marked image. This technique is extensively utilized in domains such as medical imaging, military imaging, and legal forensics, where the integrity of the original digital material is paramount.

Nowadays, three major approaches concerning RDH in JPEG images have been receiving increasing attention: (1) quantization-table-based RDH [1,2], (2) Huffman-codes-based RDH [3,4,5,6], and (3) Discrete Cosine Transform (DCT)-coefficient-based RDH [7,8,9,10,11,12,13,14,15,16].

The first approach, predicated on the alteration of quantization tables for the embedding of secret data, was introduced by [1] and subsequently refined by [2]. Although this strategy creates space for embedding and ensures a specified level of visual quality, it has resulted in a considerably substantial FSI.

The second approach embeds secret data by modifying or mapping the variable-length codes (VLCs) in the JPEG bitstream, which preserves the visual quality of the image during data embedding, albeit with a relatively limited embedding capacity (EC). In [3], a direct mapping approach was proposed, targeting the transposition of a used VLC to an unused VLC based on differing embedding bits. Utilizing the one-to-one relationship between the run/size value (RSV) and VLC, Du et al. [4] proposed an HS-based strategy that sorts RSVs according to their frequency of occurrence. While this approach can enhance the EC, it may also induce a notable increase in file size. In [5], a simulated embedding model was developed by ordering RSV occurrences to find a more effective mapping relationship for data embedding. In [6], a new VLC encoding mapping method was introduced, which utilizes a genetic algorithm to solve the mapping method and customize the VLC encoding accordingly. This method performs well in terms of file size increment compared to previous methods.

The third approach accomplishes the embedding of secret data by modifying the quantized DCT coefficients, which include alternating current (AC) and direct current (DC). Huang et al. [7] initially introduced an HS-based RDH scheme, wherein the AC coefficients possessing values of ±1 were employed to embed secret data. To maintain the reversibility of the process and make room for the data, the other AC coefficients, excluding zero-valued ones, are shifted. Hou et al. [8] formulated a new analog embedding-centric distortion function and a block selection mechanism aimed at minimizing visual distortion. The above two methods were optimized in [9], wherein a multi-faceted optimization approach was recommended, striving to pinpoint optimal decision variables among payload, FSI, and visual distortion. Xiao et al. [10] optimized the embedding distortion (ED) by choosing different embedding positions at different frequencies. Li et al. [11] proposed a novel 2D mapping strategy that categorizes the non-zero AC coefficient pairs into four distinct groups, enhancing the visual quality of the image. In [12,13,14,15,16], methods based on 2D histogram mapping were progressively developed and augmented through the optimization of distortion functions, the systematization of mapping methods, and the tailoring of adaptive mapping techniques.

In the preceding RDH methodologies employing a 2D mapping strategy for the DCT coefficients, two principal issues were discerned. The initial concern pertains to the lack of subsequent block selection from the FSI standpoint during the embedding phase; the second issue is the unavailability of the frequency ordering post data embedding, necessitating the use of an additional 63 bits to document the selected frequencies. Confronted with the challenges previously described, we were inspired to develop a new JPEG-based RDH scheme aimed at elevating the overall performance of the system. The contributions of this paper are delineated below.

We implement a dynamic frequency selection method based on a recoverable frequency order to ascertain the most suitable frequencies to embed data;We document the run length of NZACPs during their construction, facilitating the prospective estimation of file size increment and visual distortion;We group image blocks based on their metrics and adaptively prioritize the appropriate block groups for data embedding.

The remainder of this paper is structured as follows: Section 2 discusses the necessary preliminary knowledge. Section 3 details our proposed method. Experimental results are presented in Section 4, and Section 5 provides the concluding remarks.

## 2. Preliminary Knowledge

### 2.1. Overview of JPEG Compression

JPEG, widely utilized for the lossy compression of digital images, especially those from digital photography, follows a sequential process for grayscale images. This process involves three consecutive steps: Discrete Cosine Transform (DCT), quantization, and VLC assignment.

We can use the formulas below to realize the DCT transformation and its inverse procedure.
(1)$$F\left(u,v\right)=\frac{1}{4}C\left(u\right)C\left(v\right)\left[\sum_{x=0}^{7}\sum_{y=0}^{7}f\left(x,y\right)\cos\frac{(2x+1)u\pi }{16}\cos\frac{(2y+1)v\pi }{16}\right]$$
(2)$$f\left(x,y\right)=\frac{1}{4}\left[\sum_{u=0}^{7}\sum_{v=0}^{7}C\left(u\right)C\left(v\right)F\left(u,v\right)\cos\frac{(2x+1)u\pi }{16}\cos\frac{(2y+1)v\pi }{16}\right]$$
(3)$$C\left(u\right)=\left\{\begin{array}{cc} \frac{1}{\sqrt{2}}, & \mathrm{if}\;u=0\\ 1, & \mathrm{otherwise}\; \end{array}\right.$$
where f(x,y) represents the original image, which we will also denote as I(x,y). And F(u,v) stands for the unquantified DCT coefficients.

To diminish the quantity of information, each DCT coefficient value is divided by a quantization step and the quotient is rounded to the nearest integer. The magnitude of the quantization step dictates the compression ratio, with larger steps yielding more substantial compression.
(4)$$C\left(u,v\right)=\left\lfloor \frac{F(u,v)}{Q(u,v)}\right\rfloor$$
where C(u,v), Q(u,v) represent the quantized DCT coefficient and quantization value at the position of (u,v). Note that the value of Q(u,v) is related to the quality factor (QF); the larger the QF, the smaller the Q(u,v) value and the higher the visual quality of the JPEG image.

Finally, entropy coding, which is applied to generate the JPEG bitstream, has several stages. It is implemented in 8×8 blocks, and here, we provide an encoding example for a single block:**Step 1**: Scan the block in a zigzag pattern to obtain a coefficient sequence.**Step 2**: Convert the coefficient sequence to an RSV sequence. RSV is constructed for non-zero coefficients in sequences. For each non-zero coefficient, it is converted to the number of zero coefficients between the previous non-zero coefficient and current coefficient, the length of the binary representation of the coefficient, and the binary representation of the coefficient; note that the first element should not exceed 15. If there are 16 consecutive zeros, construct an RSV with the values (15,0,0). In addition, when encountering negative numbers, the representation method for the third element is to invert the highest bit of its binary representation of its absolute value.**Step 3**: Merge the first two elements in RSV into a single byte, with the top 4 bits and the bottom 4 bits representing the two elements, respectively. Due to the limitations on element values during the RSV construction process, there will be no overflow or other issues during merging.**Step 4**: The Huffman table in the JPEG Header contains a value table and a bit table. The value table corresponds one-to-one with the merged byte, and the bit table determines the length of the encoding corresponding to that byte. Based on these two tables, the byte can be converted into a Huffman code (also known as VLC in JPEG encoding).**Step 5**: Merge the obtained Huffman code with the third element of the RSV; at this point, the RSV is converted into a binary sequence; After converting all RSVs within the block, convert the entire binary sequence to hexadecimal.

The above is the entropy encoding process for a block. The complete process involves traversing and encoding all 8×8 blocks.

### 2.2. Overview of HS-Based RDH

The 1D HS-Based RDH for JPEG images is initially proposed by Huang [7]. In this scheme, the quantized DCT coefficients are divided into 8×8 blocks initially. Each block has 1 DC coefficient and 63 AC coefficients. Then, we scan the whole image in row order to obtain the sequence of blocks $\left\{B_{1},B_{2},\cdots ,B_{K}\right\}$. Each block is then converted into a coefficient sequence in a zigzag pattern, with the 63 AC coefficients designated as frequency bands $F_{i}\in \left\{F_{1},\ldots ,F_{63}\right\}$.

Note that, in this method, we have a classification of the DCT coefficients for zero-valued coefficients, which remain constant during both embedding and extraction; for coefficients with an absolute value of 1, we define them as insertable coefficients, which are used for embedding of the secret data; and for coefficients with an absolute value greater than 1, we define them as shiftable coefficients, which are used to make space for the embedding of the data and to ensure the reversibility of the method.

Based upon the conditions above, the data embedding process is mathematically articulated as follows:(5)$$C_{k}^{\prime }\left(i\right)=\left\{\begin{array}{cc} C_{k}\left(i\right)+sign\left(C_{k}\left(i\right)\right)*b, & \mathrm{if}\;|C_{k}\left(i\right)|=1\\ C_{k}\left(i\right)+sign\left(C_{k}\left(i\right)\right), & \mathrm{if}\;|C_{k}\left(i\right)|>1 \end{array}\right.$$
where $C_{k}\left(i\right)$ stands for the DCT coefficient value of the *i*th frequency band in the *k*th block. $C_{k}^{\prime }\left(i\right)$ represents the $C_{k}\left(i\right)$ with secret bit embedded and b∈{0,1} represents the secret bit. sign(·) is sign function.
(6)$$sign\left(x\right)=\left\{\begin{array}{cc} -1, & \mathrm{if}\;x<0\\ 0, & \mathrm{if}\;x=0\\ 1, & \mathrm{if}\;x>0 \end{array}\right.$$
And the extraction process is articulated as follows:(7)$$b=\left\{\begin{array}{cc} 0, & \mathrm{if}\;|C_{k}^{\prime }\left(i\right)|=1\\ 1, & \mathrm{if}\;|C_{k}^{\prime }\left(i\right)|=2 \end{array}\right.$$
(8)$$C_{k}\left(i\right)=\left\{\begin{array}{cc} \mathrm{sign}\left(C_{k}^{\prime }\left(i\right)\right), & \;\mathrm{if}\;1\leq \left|C_{k}^{\prime }\left(i\right)\right|\leq 2\\ C_{k}^{\prime }\left(i\right)-\mathrm{sign}\left(C_{k}^{\prime }\left(i\right)\right), & \;\mathrm{if}\;\left|C_{k}^{\prime }\left(i\right)\right|\geq 3 \end{array}\right.$$
Through Equations (Equation 7) and (Equation 8), the DCT coefficients with secret data can be recovered to their original values and the secret data can be extracted without error.

### 2.3. File Size Increment Table

The table, designated as hcit in [17], encapsulates the incremental changes in the entropy code’s length resulting from the data embedding process. This table is formulated using the BITS and HUFFVAL lists from the AC Huffman Table, readily extractable from the DHT segment within the JPEG Header. Consequently, the augmentation in file size for a marked image, caused by the increase of a non-zero AC coefficient in the *i*th frequency of the *k*th block can be represented as $S_{k}\left(i\right)$ and calculated by
(9)$$S_{k}\left(i\right)=\left\{\begin{array}{cc} hcit[r_{k}\left(i\right),c_{k}\left(i\right)]+1, & \mathrm{if}\;|C_{k}\left(i\right)|=2^{z}-1\\ 0, & \mathrm{others}\; \end{array}\right.$$
where $r_{k}\left(i\right)$ and $c_{k}\left(i\right)$ represent the run length of the coefficient (the number of zero coefficients in the interval between two non-zero coefficients) and the binary length of the coefficient, respectively.

### 2.4. The Laplacian Cumulative Distribution Function

He [18] proposed a scheme to approximate the distribution of all AC coefficients across frequencies using the Laplacian cumulative distribution function (CDF). This function, denoted by $F_{i}\left(x\right)$, derives from the foundational work of [19,20], and it facilitates the estimation of the AC coefficient distribution within the *i*th frequency. The CDF is delineated as follows:(10)$$F_{i}\left(x\right)=\frac{1}{2}+\frac{1}{2}\;\cdot \;sign\left(x\right)\;\cdot \;\left(1-e^{\lambda_{i}\left|x\right|}\right)$$
where $\lambda_{i}\left(>0\right)$ is the scale parameter.

Considering that AC coefficients with zero value are unaffected post-embedding, their ratio among *K* blocks can be represented as follows:(11)$$P_{Z}\left(i\right)=\frac{\sum_{k=1}^{K}\left[C_{k}\left(i\right)==0\right]}{K}$$
And the ratio can also be represented by CDF
(12)$$P_{Z}\left(i\right)=F_{i}\left(0.5\right)-F_{i}\left(-0.5\right)$$
Therefore, the scale parameter can be solved by combining Equations (Equation 11) and (Equation 12)
(13)$$\lambda_{i}=-2\;\cdot \;\ln\left[1-\frac{1}{K}\sum_{k=1}^{K}\left[C_{k}\left(i\right)==0\right]\right]$$
Once we obtain the scale parameter, the ratio of the insertable coefficient $P_{E}\left(i\right)$ and the shiftable coefficients $P_{S}\left(i\right)$ can be solved by
(14)$$\begin{array}{cc} P_{E}\left(i\right) & =\left[F_{i}\left(-0.5\right)-F_{i}\left(-1.5\right)\right]+\left[F_{i}\left(1.5\right)-F_{i}\left(0.5\right)\right]\\ \; & =e^{-\frac{1}{2}\lambda_{i}}-e^{-\frac{3}{2}\lambda_{i}}\\ P_{S}\left(i\right) & =1-P_{E}\left(i\right)-P_{Z}\left(i\right) \end{array}$$

### 2.5. Distortion Calculation

The visual distortion caused by secret data embedding we denote by ED, whose value can be represented by
(15)$$\mathrm{ED}\approx \sum_{i=1}^{M}\sum_{j=1}^{N}{∥I^{\prime }\left(i,j\right)-I\left(i,j\right)∥}^{2}$$
Based on Parseval’s Theorem, it can be concluded that
(16)$$\sum_{i=1}^{M}\sum_{j=1}^{N}{∥I^{\prime }\left(i,j\right)-I\left(i,j\right)∥}^{2}=\sum_{i=1}^{M}\sum_{j=1}^{N}{∥F^{\prime }\left(i,j\right)-F\left(i,j\right)∥}^{2}$$
And F(i,j)=C(i,j)·Q(i,j), so we can deduce that
(17)$$\mathrm{ED}\approx \sum_{k=1}^{K}\sum_{i=1}^{63}Q^{2}\left(i\right)\;\cdot \;{∥C_{k}^{\prime }\left(i\right)-C_{k}\left(i\right)∥}^{2}$$
where Q(i) represents the quantization value of the *i*th frequency band in the quantization table. Due to the quantization table used being 8×8, the summation method of Equation (Equation 17) has changed from the perspective of the entire image to 63 AC coefficients per block (DC coefficients do not carry secret data).

Since, during the data embedding process, for any non-zero AC coefficient that would be used for embedding or shifting, it changes by at most 1, we can simplify the above equation to be
(18)$$\mathrm{ED}\approx \sum_{k=1}^{K}\sum_{i=1}^{63}Q^{2}\left(i\right)$$

In conclusion, as delineated in Equation (Equation 18), we have derived a method to compute the simulated visual distortion, which is expressed as the square of the quantized value, corresponding to the modified coefficient’s position.

## 3. Proposed Method

The framework of the proposed method and its constituent processes are delineated in this section. The methodology primarily comprises two elements: dynamic frequency selection and block-based influential metrics analysis followed by grouping. Initially, we iteratively select frequencies with the help of payload and offset. NZACPs are then constructed within each block based on the frequencies identified in the initial phase. Subsequently, we estimate the ED and FSI resulting from the alteration of these coefficients. Blocks are grouped and sorted based on these two metrics, with suitable clusters of blocks being earmarked for data embedding. The comprehensive method is depicted in Figure 1.

### 3.1. Theoretical Foundation

The goal of the method is to achieve higher PSNR and lower FSI. With these objectives in mind, firstly, in block processing, we hope that when the coefficients within the block are modified, the distortion caused can be as small as possible, leading to Section 3.2; After obtaining the block order, we found that the distortion caused by modifying coefficients at different frequencies within each block is also different. Based on this, we introduce Section 3.3; after confirming the suitable frequencies, we follow the principles in Section 3.4 to construct a 2D mapping on the blocks based on the frequencies. At the same time, we noticed that, when processing according to block order, some blocks, although they cause relatively less distortion when modified, will result in a greater increase in FSI. In response, we introduce an influential model in Section 3.5 and Section 3.6, which facilitates the identification of block groups that permit data embedding with minimal distortion and FSI.

### 3.2. Block Pre-Processing

Taking a JPEG image with the shape of M×N as example, we assume that we divide this image into *K* blocks, where $K=\frac{M\times N}{8\times 8}$. The prevalence of zero coefficients in a block is indicative of its smoothness, and for a smooth block, modifying the coefficients within the block will cause less distortion. Therefore, we will first record the smoothness of the *K* blocks and use it as one of the metrics.
(19)$$MT1_{k}=\sum_{i=1}^{63}\left[C_{k}\left(i\right)==0\right]$$
where [·] stands for Iverson bracket, which assigns a value based on the truth of the proposition H it encloses. If the proposition H is true, then [H] = 1; conversely, if H is false, then [H] = 0. This bracket notation allows us to tally the non-zero AC coefficients within a block.

Subsequently, with inspiration received from [16], we will count the non-zero AC coefficients within each block and add or subtract one to each of their values. In this way, for a block, we can calculate the numerical distortion caused by all the non-zero coefficients within the block being modified by one with the Equation (Equation 18). Since we want to give higher embedding priority to blocks with less distortion, we will take the inverse of this numerical distortion as another metric.
(20)$$MT2_{k}=\sum_{i=1}^{63}\left[C_{k}\left(i\right)\neq 0\right]\;\cdot \;Q^{2}\left(i\right)$$

Based on the above, we introduce the metric MT, defined as the sum of MT1 and 1/MT2 from Equations (Equation 19) and (Equation 20). It indicates that blocks with higher MT values are smoother and result in less distortion during the modification of coefficients. Therefore, the set of blocks $\left\{B_{1},\cdots ,B_{k},\cdots ,B_{K}\right\}$ can be arranged in descending order based on MT to yield $\left\{\tilde{B_{1}},\cdots ,\tilde{B_{k}},\cdots ,\tilde{B_{K}}\right\}$.
(21)$$MT_{k}=MT1_{k}+\frac{1}{MT2_{k}}$$

### 3.3. Dynamic Frequency Selection

After the completion of block pre-processing, a selection of frequencies becomes imperative. There are two main reasons for doing so: the first is that DCT coefficients located in the high-frequency domain correspond to larger quantization coefficients, which can lead to a higher ED; the second is that it is possible that most of the coefficients contained at that frequency cannot be used for data embedding, like coefficients with an absolute value greater than 1 in the HS-based method, which can only be used to make room for data embedding rather than embedding data.

Inspired by [8,18], we sort frequencies based on the unit distortion of different frequencies, and the metric UD(i) is calculated as follows, in conjunction with Equation (Equation 18):(22)$$UD\left(i\right)=\left(\frac{1}{2}+\frac{P_{S}\left(i\right)}{P_{E}\left(i\right)}\right)\;\cdot \;Q^{2}\left(i\right)$$
where $P_{E}\left(i\right)$ and $P_{S}\left(i\right)$ denote the ratio of insertable and shiftable coefficients in the *i*th band of *K* blocks, respectively. The detailed calculation process concerning these two metrics is shown in Section 2.4. Note that the metrics utilized to establish the frequency order remain invariant prior to and subsequent to the data embedding process, that is, the frequency order can be reconstructed post-data embedding without necessitating additional data.

Upon sorting the frequencies, a dynamic selection is made using an offset, which operates under the principle that selecting a greater number of frequencies correlates with the use of fewer blocks, and conversely. It is observed that, for certain images, a selection of less distorted frequencies and a higher number of blocks yields improved performance. Conversely, other images benefit from selecting more frequencies and fewer blocks. Consequently, our selection strategy is guided by the order of frequencies, with the requirement that the total EC must surpass the combined value of the payload *P* and the offset *O*.
(23)$$\sum_{i=1}^{r}\mathrm{EC}\left(i\right)\geq P+O$$
(24)$$\mathrm{EC}\left(i\right)=\sum_{k=1}^{K}\left[|C_{k}\left(i\right)|==1\right]$$
from the Equations (Equation 23) and (Equation 24), we can obtain the optimal frequency set $F^{*}=\left\{F_{1}^{*},\cdots ,F_{r}^{*}\right\}$

This part represents one cycle of selecting the frequencies, after simulating data embedding, we can compute the total distortion *T*. And *T* is numerically equal to ED from Equation (Equation 15). In the actual embedding process, we loop the offset set to obtain the minimum total distortion $T^{*}$. The detailed process is shown in Section 3.7.

### 3.4. Two-Dimensional Mapping Generation

The 2D mapping uses the same strategy as Li [11], defining the NZACP within each block into four categories, namely A, B, C, and D, respectively. Under these, the process of modifying the coefficients in NZACPs for data embedding is 2D HS. The mapping strategy is shown in Figure 2a, with the first quadrant serving as an example.

The acquired coefficient pairs are those that ultimately complete the embedding of the secret data, each according to its specific type, whilst ensuring the image’s reversibility. The comprehensive data embedding procedure is delineated as follows:**Type A**: They are defined by the set {(x=±1,y=±1)}. When embedding data, they stay the same when they encounter a 0. If they encounter a 1, they look back one place, and the coefficient pairs are shifted 1 place on the x-axis if the next place is a 0, and 1 place on the y-axis if the next place is a 1.**Type B**: They fall within {(x=±1,y≠±1)} or {(x≠±1,y=±1)}. When embedding data, a 0 is shifted 1 bit on the x-axis or y-axis when encountered; if a 1 is encountered it is shifted 1 bit along the diagonal direction.**Type C**: They are categorized as {(x=±2,y=±2)}. When embedding data, they stay the same when they encounter a 0 as Type A; if a 1 is encountered it is shifted 1 bit along the diagonal direction as Type B.**Type D**: All remaining coefficient pairs are classified as Type D and are solely shifted diagonally. This shifting is primarily utilized to maintain reversibility and does not embed secret data.

In the process of constructing NZACPs, we document not only the type of each NZACP, but also the run length and the corresponding frequency associated with both coefficients within each pair. This step helps us later to compute the ED and FSI for each block. The restoration process can be easily accomplished through a few steps. From Figure 2b, we can easily identify their types, extract corresponding data, and follow the principles to shift them back to their original positions.

### 3.5. Influential Model Construction

Utilizing $\left\{\tilde{B_{1}},\cdots ,\tilde{B_{k}},\cdots ,\tilde{B_{K}}\right\}$, $F^{*}$ and the mapping strategy described in Section 3.4, we can construct an influential model for every block. This model facilitates the computation of ED and FSI caused by available coefficient pairs for each block, after embedding or shifting. Moreover, it records the EC of each block. An example of the construction and computational procedures is provided, with details presented in Figure 3.

Figure 3 illustrates a random 8×8 block for computation. Initial processing involves a zigzag traversal of this block to distill the non-zero coefficients, concurrently documenting their run lengths. Subsequently, we pair two adjacent non-zero coefficients. Take the third coefficient pair as an example. This coefficient pair corresponds to **Type B**, i.e., it pans one unit horizontally or vertically or one unit diagonally depending on the secret data to be embedded. Predicated on the trajectory of the movement, we can deduce its EC, and when amalgamated with the principles from Equations (Equation 9) and (Equation 18), we are equipped to compute the ED and FSI. Thus, for each block $\tilde{B_{k}}$, we are capable of calculating its respective ${\mathrm{EC}}_{k},{\mathrm{ED}}_{k},{\mathrm{FSI}}_{k}$.

### 3.6. Adaptive Block Grouping

Guided by the model mentioned in Section 3.5, each block $\tilde{B_{k}}$ is characterized by the metrics ${\mathrm{ED}}_{k},{\mathrm{FSI}}_{k},{\mathrm{EC}}_{k}$. For illustrative purposes, consider a 512×512 image as an example, which yields 4096 blocks; these are then segregated into 64 groups, denoted as $group_{m},m=1,\cdots ,64$, based on the aforementioned metrics. This categorization results in the parameters $group\_{\mathrm{ED}}_{m}$, $group\_{\mathrm{FSI}}_{m}$, and $group\_{\mathrm{EC}}_{m}$. Subsequently, these 64 groups are ordered in an ascending sequence according to the influence factor infl, computed as
(25)$$infl_{m}=\mathrm{normalize}\left(group\_{\mathrm{ED}}_{m}\right)+\mathrm{normalize}\left(group\_{\mathrm{FSI}}_{m}\right)$$
The selection process involves iterating over all groups from the beginning, continuing until the cumulative embedding capacity $\sum_{m=1}^{M}group\_{\mathrm{EC}}_{m}\geq P$. Ultimately, this yields *M* block groups ready for data embedding.

### 3.7. Data Embedding and Extracting

The data embedding procedure is delineated with precision as follows:**Step 1**: Decode the original JPEG bitstream to obtain the quantized DCT coefficient matrix and the quantization table. Initialize the total distortion $T^{*}$ to positive infinity. Arrange all DCT blocks by their own MT in descending order.**Step 2**: Compute the unit distortion UD for all frequency bands, considering Equation (Equation 22) to set their initial priorities.**Step 3**: Select $F^{*}=\left\{F_{1}^{*},\cdots ,F_{r}^{*}\right\}$ frequencies based on the payload *P* and offset *O*. Apply the 2D mapping strategy to construct NZACPs on the sorted *K* blocks and $F^{*}$, and compute the ${\mathrm{ED}}_{k},{\mathrm{FSI}}_{k},{\mathrm{EC}}_{k}$ of each block.**Step 4**: Group the blocks by ${\mathrm{ED}}_{k},{\mathrm{FSI}}_{k},{\mathrm{EC}}_{k}$. Sort and select *M* block groups for data embedding.**Step 5**: Simulate the embedding of secret data and record the total distortion *T*. If $T<T^{*}$, then $T^{*}=T$, and keep record of the auxiliary data for this case. If all the *O* have been traversed, then go to **Step 6**, otherwise go back to **Step 3**.**Step 6**: Sequentially embed secret data in the optimal frequency band set $F^{*}$ and *M* selected block groups. Then, encode the DCT coefficients with secret data as the marked image.

Notably, it is imperative to document the length of the optimal frequency band set $L^{*}$, the chosen *M* block groups, and the payload *P*, which occupy 6 bits, 64 bits, and $\log_{2}P$ bits, respectively. These elements are sequentially embedded into the reserved space within the JPEG Header as auxiliary data.

Data extraction and image restoration processes are executed with ease.

**Step 1**: Extract the auxiliary data $L^{*}$, *M*, and *P* from the reserved space within the JPEG Header.**Step 2**: Recover the optimal frequency band set $F^{*}$ with Equation (Equation 22) and $L^{*}$. The equation can restore the order of frequencies and $F^{*}$ is the first $L^{*}$ frequencies in the order.**Step 3**: Rearrange the block order with MT. As MT remains unchanged after data embedding, the block order can be directly restored.**Step 4**: Reconstruct the NZACPs by utilizing $F^{*}$ in conjunction with the *M* block groups. Then, from Figure 2b, we can easily identify the type of NZACP with secret data, as well as ascertain the shifting direction for image recovery and the extraction of the corresponding data.**Step 5**: Sequentially extract the secret data and recover DCT coefficients via an inverse 2D mapping shift. Then, encode the restored DCT coefficients to obtain the original image.

## 4. Experimental Results

This section begins by detailing the experimental settings and the selection of image datasets. Subsequent comparisons are drawn between our methodology and both one-dimensional as well as two-dimensional schemes. The concluding portion assesses our method’s comprehensive performance relative to several contemporary techniques, with the findings systematically presented.

### 4.1. Experimental Setup

We conducted a series of experiments to showcase the advantages of our scheme over USC-SIPI imageset, CVG-UGR image database, and Bossbase v1.01 image set. The first two sets include multiple commonly used testing images such as Lena, Baboon, Goldhill, etc. The last set consists of numerous natural images. All experimental images were resized to 512×512 and re-compressed into JPEG format with different quality factors.

The overall performance is evaluated using two primary metrics: visual quality, assessed by PSNR, and FSI. These metrics provide a comprehensive evaluation of the experimental results. The FSI is determined by comparing the storage space occupied by a JPEG image before and after the data embedding process. Concurrently, the PSNR metric is utilized to appraise the visual quality of the image.
(26)$$\mathrm{PSNR}=10\;\cdot \;\log_{10}\left(\frac{{\mathrm{MAX}}_{I}^{2}}{\mathrm{MSE}}\right)dB$$
where ${\mathrm{MAX}}_{I}$ represents the maximum achievable pixel value in the image, while the mean square error (MSE) of the spatial image is computed using the equation below:(27)$$\mathrm{MSE}=\frac{1}{M\times N}\sum_{i=1}^{M}\sum_{j=1}^{N}\left|\right|I^{\prime }\left(i,j\right)-I\left(i,j\right){\left|\right|}^{2}$$
where $I^{\prime }$ and *I* are the marked image and original image, respectively, and M,N denote the shape of the JPEG image.

### 4.2. Assessment of Visual Quality and File Size Increment

A series of experiments were performed to assess the visual quality and file size increment in this section. The proposed scheme demonstrates the capability to preserve high visual imperceptibility post data embedding, applicable to images regardless of their textured or smooth characteristics. Furthermore, the scheme was bench-marked against a variety of JPEG RDH schemes, which are categorizable into two types: schemes based on 1D HS and those founded on 2D HS.

#### 4.2.1. Evaluating against One-Dimensional Methodologies

We initially compared our scheme with four JPEG RDH schemes based on 1D HS, including Huang [7], Hou [8], Yin [9], He [17]. Five quintessential test images from the USC-SIPI image set were employed to appraise the PSNR and FSI efficacy. The resultant experimental data are shown in Table 1. Note that the bold data in Table 1 represent the best performance in their respective experimental settings. And this rule also applies to Table 2 and Table 3.

From Table 1, the superiority of our method over the 1D approaches is evident, achieving a higher PSNR and a smaller FSI in most scenarios. For instance, our approach exhibits a PSNR for the Lena image at a capacity of 5000 bits capacity that is 0.11% higher than the nearest competitor, while simultaneously reducing the FSI by 7.19% compared to the lowest FSI reported by other methods. This marked improvement stems from our strategic block selection, guided by ${\mathrm{EC}}_{k}$, ${\mathrm{ED}}_{k}$, and ${\mathrm{FSI}}_{k}$, which leads to block groups that achieve a lower FSI and a higher PSNR. Additionally, our dynamic frequency selection method has pinpointed a frequency set with minimized unit embedding distortion.

#### 4.2.2. Evaluating against Two-Dimensional Methodologies

Furthermore, to demonstrate the superiority of our scheme, comparisons were drawn with counterparts based on 2D HS, including Li-N [11] and Li-F [12]. These comparisons were conducted by adjusting the quality factors to 70, 80, and 90, respectively, across five classical images sourced from the USC-SIPI image set and the CVG-UGR image database. The outcomes are exhibited in Table 2 and Table 3.

From Table 2 and Table 3, it can be observed that our method is also superior to several existing methods of the same kind in various aspects. For FSI, our method consistently has better performances. At a QF of 80 for 6000 bits, our FSI values are 6376 bits for Lena, 5912 bits for Peppers, 5528 bits for Tiffany, 7368 bits for Goldhill, and 5640 bits for Splash. These figures are lower when compared to the other two methods. In the context of PSNR, our method also shows a clear advantage. At a QF of 70 for 6000 bits, our PSNR values are 47.281 dB for Lena, 48.030 dB for Peppers, 47.897 dB for Tiffany, 46.082 dB for Goldhill, and 48.298 dB for Splash, surpassing the corresponding figures from the other two methods.

### 4.3. Evaluating against the State of the Arts

To furnish additional insights, we expanded our experimental regimen to encompass the BOSSbase v1.01 database, thus affirming the wide applicability and robustness of our proposed scheme. A set of 50 images was arbitrarily chosen from the database for testing. All experimental images underwent conversion to JPEG format using three QFs (70, 80, 90), and eight different payloads were used to evaluate the corresponding performances. Comparative experiments were conducted against three established schemes: Huang [7], He [17], Li-N [11], Li-F [12], focusing on PSNR and FSI. The findings from these experiments are exhibited in Figure 4 and Figure 5.

From Figure 4 and Figure 5, it can be seen that, across all the experimental environments we established, our method consistently outperforms others in terms of PSNR and FSI.

In addition, we will further compare the running time of different methods. We record the experimental times in Figure 4 and Figure 5 and calculate the average running time of each method uniformly. The results are shown in Table 4.

Combined with the discussion above, Table 4 reveals that the Huang [7] method, the earliest developed, is the fastest. It directly selects the embedding location for secret data based on preliminary extensive findings; however, it lags in terms of PSNR and FSI performance. The He [17] method demonstrates good performance in terms of PSNR; however, when considering FSI, it falls short in comparison to methods utilizing 2D mapping. Additionally, its average running time is higher than that of other methods. The Li-N [11] method is marginally faster than ours, yet it under-performs in terms of PSNR and FSI. On the other hand, the Li-F [12] method shows some improvement over the Li-N [11] method in terms of performance but it is almost four times slower than ours, and its performance also lags behind our method.

## 5. Conclusions

Confronting the issue of balancing visual quality with file size increment in JPEG RDH, this paper unveils a new scheme that utilizes 2D mapping. Our scheme treats blocks as discrete units, and block groups are chosen based on the assessment of their influential metrics. It also incorporates the Laplacian cumulative distribution function into the unit distortion computation for frequency selection. The experimental results clearly indicate that our proposed scheme surpasses multiple contemporary JPEG RDH methods in visual quality and file size increment.

## Figures and Tables

**Figure 1 entropy-26-00301-f001:**
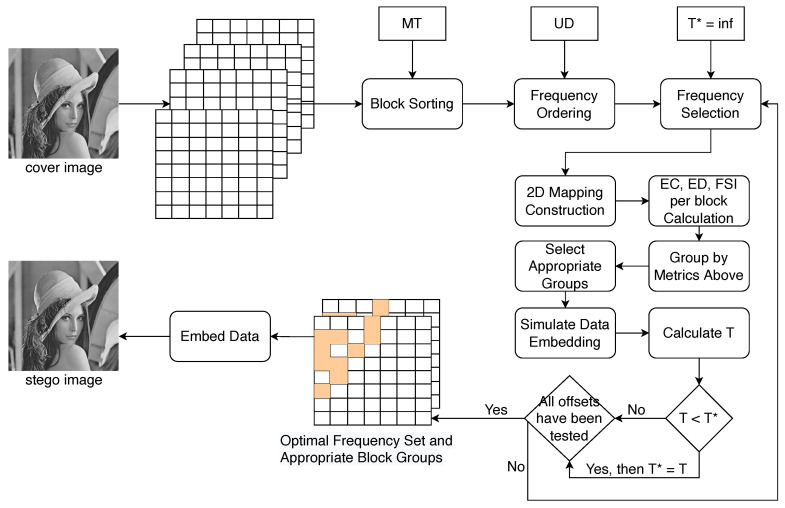
Illustration of the proposed scheme, using the gray Lena JPEG image as an example.

**Figure 2 entropy-26-00301-f002:**
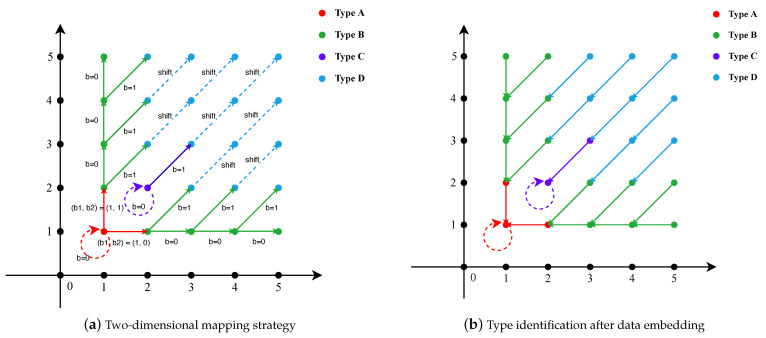
2D mapping strategy and type identification after data embedding.

**Figure 3 entropy-26-00301-f003:**
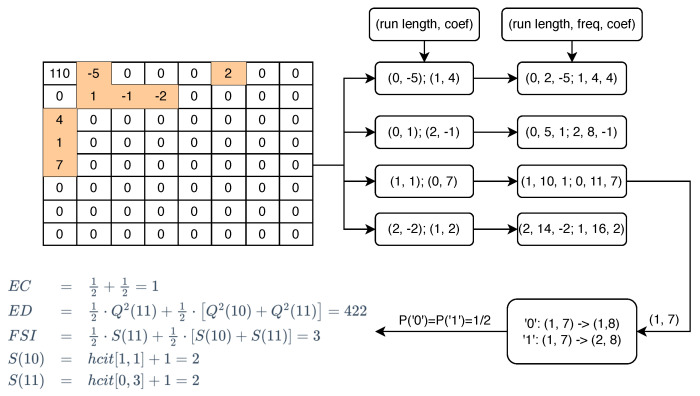
Example of influential model with computational procedures.

**Figure 4 entropy-26-00301-f004:**
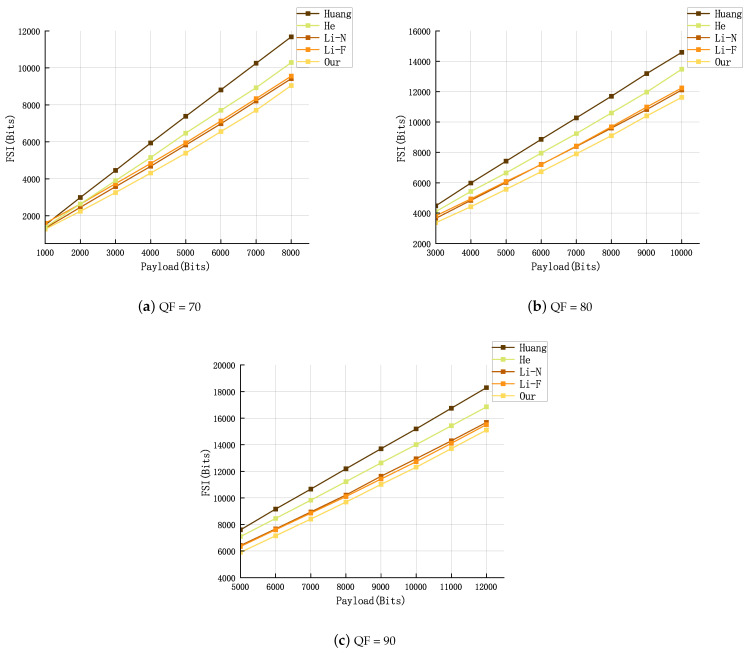
FSI comparison for three quality factors (QF = 70, 80, 90) [7,11,12,17].

**Figure 5 entropy-26-00301-f005:**
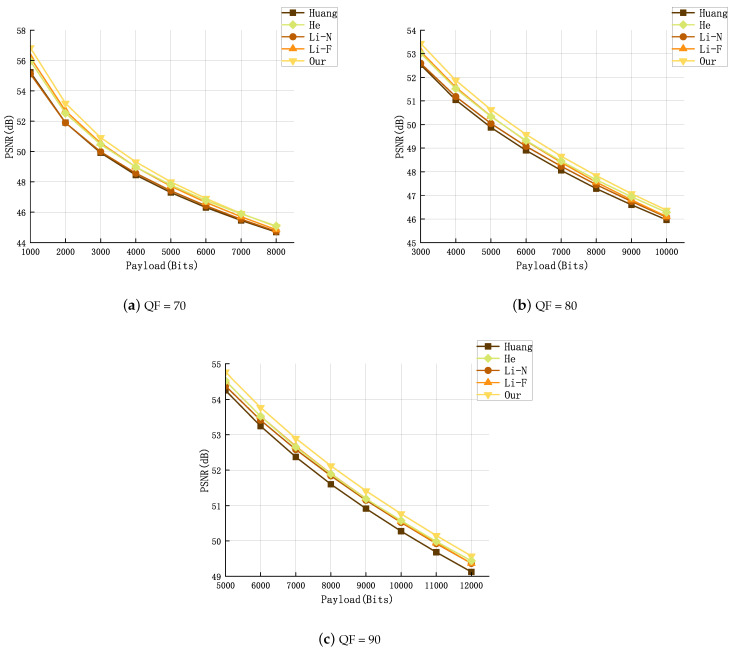
PSNR comparison for three quality factors (QF = 70, 80, 90) [7,11,12,17].

**Table 1 entropy-26-00301-t001:** One-Dimensional JPEG RDH Schemes Comparison under QF = 90.

Images	Metric	Huang [7]		Hou [8]		Yin [9]		He [17]		Our	
		**5000**	**10,000**	**5000**	**10,000**	**5000**	**10,000**	**5000**	**10,000**	**5000**	**10,000**
Lena	PSNR	54.393	50.989	54.945	51.365	55.561	51.693	55.418	51.913	**55.623**	**52.053**
	FSI	7624	14,328	6560	13,464	5672	11,624	6392	12,080	**5264**	**9992**
Baboon	PSNR	49.236	45.323	49.636	45.330	49.963	45.664	50.493	46.225	**50.836**	**46.550**
	FSI	8360	17,008	7536	17,000	7712	16,008	7544	16,736	**7328**	**14,416**
Tiffany	PSNR	52.560	49.034	53.335	49.578	53.883	50.002	54.170	50.810	**54.348**	**51.005**
	FSI	7696	14,968	6328	14,032	6160	12,856	6560	11,688	**5608**	**10,632**
Peppers	PSNR	52.975	49.169	54.047	50.117	54.608	50.472	54.859	51.393	**55.092**	**51.502**
	FSI	8048	14,848	6872	14,376	6384	12,824	6016	12,072	**5408**	**10,688**
Couple	PSNR	51.623	47.772	52.702	48.274	53.240	48.818	53.806	**49.835**	**53.871**	49.700
	FSI	7328	15,208	6640	14,728	6408	13,112	6408	13,520	**5928**	**12,736**

**Table 2 entropy-26-00301-t002:** Two-Dimensional JPEG RDH Schemes PSNR Comparison under Different QFs.

Image	Scheme	QF = 70			QF = 80			QF = 90		
		6000	9000	12,000	6000	9000	12,000	6000	9000	12,000
Lena	Li-N [11]	46.864	44.178	41.684	50.205	47.717	45.594	54.503	52.381	50.806
	Li-F [12]	46.894	44.213	41.635	50.457	47.745	45.637	54.571	52.479	50.797
	Our	**47.281**	**44.311**	**41.743**	**50.633**	**47.977**	**45.758**	**54.806**	**52.646**	**50.933**
Peppers	Li-N [11]	47.282	44.773	42.626	50.094	47.788	46.031	53.978	51.801	50.151
	Li-F [12]	47.563	44.943	42.650	50.642	48.208	46.037	53.893	51.667	50.097
	Our	**48.030**	**45.262**	**42.832**	**50.683**	**48.354**	**46.428**	**54.219**	**52.100**	**50.446**
Tiffany	Li-N [11]	47.347	44.700	42.582	49.842	47.605	45.795	53.163	51.200	49.703
	Li-F [12]	47.569	44.888	42.592	50.180	47.693	45.743	53.111	51.165	49.582
	Our	**47.897**	**45.005**	**42.770**	**50.393**	**48.027**	**46.046**	**53.452**	**51.525**	**49.985**
Goldhill	Li-N [11]	45.461	43.318	41.722	47.697	45.549	43.985	51.746	49.400	47.622
	Li-F [12]	45.737	43.579	41.790	47.916	45.725	44.032	51.733	49.382	47.571
	Our	**46.082**	**43.842**	**42.065**	**48.270**	**46.034**	**44.426**	**52.143**	**49.623**	**47.852**
Splash	Li-N [11]	47.803	45.591	43.418	50.198	48.262	46.607	53.511	51.670	50.272
	Li-F [12]	47.819	45.691	43.571	50.428	48.365	46.835	53.647	51.669	50.231
	Our	**48.298**	**45.692**	**43.673**	**50.690**	**48.611**	**46.842**	**54.061**	**52.043**	**50.622**

**Table 3 entropy-26-00301-t003:** Two-Dimensional JPEG RDH Schemes FSI Comparison under Different QFs.

Image	Scheme	QF = 70			QF = 80			QF = 90		
		6000	9000	12,000	6000	9000	12,000	6000	9000	12,000
Lena	Li-N [11]	6536	10,304	14,552	6792	10,016	13,600	6688	9512	13,064
	Li-F [12]	7056	10,512	14,824	6384	10,256	13,800	6536	9640	12,984
	Our	**6488**	**10,096**	**14,528**	**6376**	**9560**	**13,360**	**6232**	**8936**	**12,824**
Peppers	Li-N [11]	6632	9864	13,312	6712	9880	12,880	6816	9944	13,152
	Li-F [12]	6808	10,368	13,576	6144	8864	13,400	6784	9560	12,792
	Our	**6088**	**9160**	**13,240**	**5912**	**8816**	**12,224**	**6440**	**9328**	**12,224**
Tiffany	Li-N [11]	6208	9624	13,464	6184	9200	12,848	7240	10,112	13,752
	Li-F [12]	6696	9944	13,896	5712	9456	13,192	6952	9984	13,200
	Our	**5976**	**9552**	**13,288**	**5528**	**8760**	**11,888**	**6552**	**9776**	**13,176**
Goldhill	Li-N [11]	7512	10,664	14,520	7808	11,544	15,192	7744	12,168	16,472
	Li-F [12]	7312	10,728	14,312	8056	11,496	15,256	7976	12,352	16,424
	Our	**6664**	**9952**	**13,752**	**7368**	**10,856**	**14,480**	**7728**	**12,024**	**16,384**
Splash	Li-N [11]	**5520**	**8480**	12,120	6232	8776	11,648	7264	10,600	13,832
	Li-F [12]	6056	8904	12,488	5984	9440	12,264	6728	10,528	14,280
	Our	**5520**	8552	**12,072**	**5640**	**8648**	**11,456**	**6688**	**9912**	**12,968**

**Table 4 entropy-26-00301-t004:** Average running time comparison in BOSSbase v1.01 database under different QFs.

QF	Average Running Time/s				
	Huang [7]	He [17]	Li-N [11]	Li-F [12]	Our
70	0.04	28.42	3.02	12.61	3.19
80	0.05	27.20	3.06	14.21	3.42
90	0.05	23.46	3.04	16.29	3.93

## Data Availability

The data presented in this study are available on request from the corresponding author.

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
