# Peer review of "Influential Metrics Estimation and Dynamic Frequency Selection Based on Two-Dimensional Mapping for JPEG-Reversible Data Hiding"

_entropy, 2024, doi:10.3390/e26040301_

Round 1
Reviewer 1 Report
Comments and Suggestions for Authors
The manuscript presents a new scheme for JPEG Reversible Data Hiding (RDH) aimed at mitigating the trade-off between visual quality and file size increment (FSI) typically encountered in existing RDH methods. The manuscript outlines the methodology, emphasizing the incorporation of a dynamic frequency selection strategy and a Two Dimensional (2D) mapping approach to address this challenge. Experimentation is reported, demonstrating the superiority of the proposed scheme over other existing techniques in terms of both visual quality, measured by Peak Signal-to-Noise Ratio (PSNR), and optimized FSI.
Strengths:
- Innovative approach: The manuscript introduces a novel scheme for JPEG RDH, incorporating a dynamic frequency selection strategy and a 2D mapping approach to address the trade-off between visual quality and file size increment. This innovative methodology demonstrates the authors' efforts to advance the state-of-the-art in RDH techniques.
- Comprehensive experimentation: Quite extensive experimentation is conducted to evaluate the efficacy of the proposed scheme, demonstrating consistent superiority over multiple contemporary JPEG RDH methods in terms of visual quality and file size increment. The inclusion of empirical evidence enhances the credibility of the research findings.
- Clear presentation: The manuscript provides a clear overview of the proposed scheme and its contributions to the field of JPEG RDH. The methodology and experimental results make it easy for readers to grasp the key aspects of the research.
Weaknesses:
- Theoretical foundation: While the manuscript describes the methodology and experimental results in detail, it lacks a thorough discussion of the theoretical foundation underlying the proposed scheme. Providing insights into the theoretical principles guiding the design of the scheme would enhance the understanding of its underlying mechanisms and rationale.
- Comparative analysis: Although the experimental results demonstrate the superiority of the proposed scheme over existing techniques, a comparative analysis with other state-of-the-art RDH methods would provide additional context and insights into the relative strengths and weaknesses of the proposed approach. Including such analysis would strengthen the manuscript's contributions and facilitate a more nuanced interpretation of the experimental results.
- Lack of Discussion on Security Issues: One notable weakness of the manuscript is the absence of discussion about potential security issues inherent in the proposed scheme. Specifically, there is no mention of the security of the embedded data, such as the protection of secret keys or the resilience against attacks aimed at extracting the embedded message without knowledge of the secret keys (if any). Addressing these security concerns would enhance the applicability of the proposed scheme in real-world scenarios.
Overall, the manuscript presents a promising contribution to the field of JPEG RDH, offering an innovative approach to address the trade-off between visual quality and file size increment. While the methodology is well-presented, further elaboration on the theoretical foundation, comparative analysis, and security analysis would strengthen the manuscript's contributions and enhance its impact in the research community.
Detailed comments for revisions are provided below:
1. Theoretical Foundation:
The manuscript presents a detailed methodology and experimental results, which provide valuable insights into the practical implementation and performance of the proposed scheme. However, to further enhance the understanding of the proposed approach, it would be beneficial to include a discussion on the theoretical foundation underlying the scheme. This discussion could encompass the fundamental principles, algorithms, and mathematical models that inform the design and operation of the scheme. By elucidating the theoretical underpinnings, readers will gain a deeper comprehension of the underlying mechanisms and rationale driving the proposed methodology. Additionally, discussing the theoretical basis would contribute to the scholarly rigor of the manuscript and enrich its academic value.
2. Comparative analysis:
While the experimental results demonstrate the superiority of the proposed scheme over existing techniques, the selection of the methods for the comparison in Section 4.3 is unclear. In particular, the authors have selected the method by Huang et al. [7] for these experiments, while it is not the best one among the ones compared in Table 1. Why do not compare with He et al. [17] which performs better thatn Huang et al. [7]?
In addition, the comparative analysis could encompass a comprehensive assessment of other key metrics such as computational complexity. By enhancing the comparative analysis, the manuscript can highlight the relative strengths and weaknesses of the proposed approach, offering readers a more nuanced understanding of its contributions to the field. Additionally, discussing the comparative performance of the proposed scheme would strengthen the manuscript's scientific rigor and facilitate a more comprehensive interpretation of the experimental results.
3. Discussion on security issues:
Addressing potential security concerns is essential for ensuring the robustness and applicability of the proposed scheme in real-world scenarios. Therefore, I recommend including a discussion on security issues inherent in the proposed scheme. Specifically, the manuscript should address aspects such as the security of embedded data, the protection of secret keys (if utilized), and the resilience against potential attacks aimed at extracting the embedded message without knowledge of the secret keys. By addressing security concerns, the manuscript can enhance the trustworthiness and practical relevance of the proposed scheme, thereby strengthening its potential for real-world applications.
4. Future research directions:
Please suggest some directions for future research.
Minor comments:
a) Please expand the acronyms the first time they are used, both in the abstract and the main text, to improve readability and comprehension for readers.
b) The text in Figure 1 is too small to be read, even with a large zoom. Consider enlarging the text or providing a higher resolution version of the figure to ensure clarity and readability.
c) Figures 4 and 5 are too small and cannot be viewed correctly without zooming extensively. Consider resizing or providing higher resolution versions of these figures to ensure they are easily interpretable by readers.
Comments on the Quality of English Language
Here are some corrections for sentences in the manuscript:
-
However, while existing RDH methods adeptly manage visual distortion caused by embedded data,
- Correction: However, while existing RDH methods adaptly manage the visual distortion caused by embedded data,
-
In rectifying this oversight, we’ve designed a new JPEG RDH scheme that deal with all influential metrics during the embedding phase
- Correction: In rectifying this oversight, we have designed a new JPEG RDH scheme that deals with all influential metrics during the embedding phase
-
the others AC coefficients excluding zero-valued are shifted.
- Correction: the other AC coefficients, excluding zero-valued ones, are shifted
-
Xiao et al. [10] optimizes the embedding distortion (ED)
- Correction: Xiao et al. [10] optimize the embedding distortion (ED)
-
In [12–16], such methods based on 2D histogram mapping are progressively evolved and augmented through the optimization of distortion functions,
- Correction: In [12–16], methods based on 2D histogram mapping are progressively evolved and augmented through the optimization of distortion functions
-
If the proposition P is ture, then
- Correction: If the proposition P is true, then
-
This way for a block, we can get the numerical distortion caused by all the non-zero coefficients
- Correction: In this way, for a block, we can calculate the numerical distortion caused by all the non-zero coefficients
-
Inspired by [8] and [18], we will sort frequencies based on the unit distortion
- Correction: Inspired by [8] and [18], we sort frequencies based on the unit distortion
Reviewer 2 Report
Comments and Suggestions for Authors
Overall, the manuscript is well-written. However, I will highlight some points here for you to consider:
* When explaining the parameters of equation (4), C(u, v) was repeated representing two different quantities. Was this intentional or a typo?
* the idea of "entropy coding" was mentioned quickly in section 2.1. I guess a bit of explanation will add some depth and complete the picture. If you can describe the JPEG compression algorithm in clear steps, it will be even better!
* at the beginning of section 2.2, you mentioned "This type of RDH ..." . I am wondering what this refers to.
* In section 2.2, in your sentence "63 AC coefficients...", what does AC refer to?
* the terms b, C' are not defined in equation (5). This is important for the reader to understand how the extraction process works.
* in section 2.3, I feel that the last sentence "The coefficient in the i th frequency, kth block.", is either redundant or improperly cut.
* Equation 17 uses 63 as the upper bound for the summation over i. Could you please explain why? if this is related to the 63 AC coefficients, please elaborate.
* can you use a higher resolution for figure 1?
* in describing the parameters of equation (19), you mentioned "where [·] stands for Iverson bracket. If the proposition P is ture, then [P] = 1; conversely, if P is false, then [P] = 0." those are not related to the equation you have given.
* equation (21) is not described nor related to the previous equations/steps.
* after equation (24), you mentioned "T ∗" and "T" in two consecutive sentences. are those the same or do you mean something else? please try to keep your symbols consistent.
* in section 3.6, you described the embedding process in detail using clear steps. However, the data extraction was explained in a very shallow way. Please use detailed steps here too.
* Please use larger images for figures 4 and 5.
* The experimental results don't show an outstanding performance compared to other methods. your claim of reducing the file size increment in JPEG RDH was not supported by the results.
Round 2
Reviewer 2 Report
Comments and Suggestions for Authors
All comments have been addressed in the reviewed version.